# Dynamic Changes in the Nutrient Digestibility, Rumen Fermentation, Serum Parameters of Perinatal Ewes and Their Relationship with Rumen Microbiota

**DOI:** 10.3390/ani14162344

**Published:** 2024-08-14

**Authors:** Jiaxin Chen, Siwei Wang, Xuejiao Yin, Chunhui Duan, Jinhui Li, Yueqin Liu, Yingjie Zhang

**Affiliations:** 1College of Animal Science and Technology, Hebei Agricultural University, Baoding 071000, China; chenjiaxin1226@163.com (J.C.); wangsiwei_8999@163.com (S.W.); duanchh211@126.com (C.D.); lijinhui80232020@163.com (J.L.); liuyueqin66@126.com (Y.L.); 2Institute of Cereal and Oil Crops, Hebei Key Laboratory of Crop Cultivation Physiology and Green Production, Hebei Academy of Agriculture and Forestry Sciences, Shijiazhuang 050035, China; 3College of Animal Science and Technology, Hebei Normal University of Science and Technology, Qinhuangdao 066004, China; bdyinxuejiao@foxmail.com

**Keywords:** sheep, perinatal period, dry matter intake, nutrient digestibility, serum parameters, rumen microbiota

## Abstract

**Simple Summary:**

Changes in physiological and biochemical parameters are crucial for the reproductive performance and health of perinatal ewes. However, dynamic changes in feed utilization, rumen fermentation, and serum biochemical indexes of perinatal ewes are unclear. This study aims to explore dynamic changes in dry matter intake, nutrient digestibility, rumen fermentation, and serum biochemical indexes in perinatal ewes and their relationship with rumen microbiota. We found that the dry matter intake and glucose gradually decreased during prepartum and increased during postpartum. The digestibility of dry matter, crude protein, and acid detergent fiber increased before lambing, and decreased on day 3 after lambing. The concentrations of acetate, propionate, and butyrate gradually decreased before lambing and increased after lambing. The rumen microbiota composition was different in perinatal ewes, and the changes in dry matter intake, serum glucose, acetate, and propionate were related to rumen bacteria (*g_Anaerovibrio*, *g_Lachnobacterium*, *g_Schwartzia* and *g_Bacillus*). The results provide a basis for the regulation of physiological and biochemical parameters of perinatal ewes by rumen microbiota.

**Abstract:**

Changes in physiological and biochemical parameters are crucial for the reproductive performance and health of perinatal ewes. This study investigated the temporal variations in feed intake, nutrient digestibility, serum parameters, and ruminal fermentation on days 21, 14, and 7 before lambing (Q21, Q14, and Q7) and days 3, 7, and 14 after lambing (H3, H7, and H14). The results showed that dry matter intake (DMI) and glucose (Glu) gradually decreased (*p* < 0.05) before lambing and increased (*p* < 0.05) after lambing. The digestibility of dry matter (DMD), crude protein (CPD), and acid detergent fiber (ADFD) increased (*p* < 0.05) before lambing, then decreased (*p* < 0.05) on day H3, and then increased (*p* < 0.05) on day H14. The rumen pH, NH_3_-N, and triglycerides (TG) gradually increased (*p* < 0.05) before lambing and were higher (*p* < 0.05) on day Q7 than after lambing. The concentrations of acetate, butyrate, and total volatile fatty acids (T-VFA) were lower (*p* < 0.05) on day Q7 than those on days Q21 and Q14, then increased (*p* < 0.05) after lambing. Total cholesterol (TC), high-density lipoprotein cholesterol (HDL-C), and low-density lipoprotein cholesterol (LDL-C) concentrations gradually decreased (*p* < 0.05) in perinatal ewes. BHBA and NEFA concentrations were lower (*p* < 0.05) on day Q21 than those from days Q14 to H14. The rumen microbiota compositions were different (*p* < 0.05) in perinatal ewes, and *g_Anaerovibrio*, *g_Lachnobacterium*, and *g_Schwartzia* were positively correlated (*p* < 0.05) with DMI, Glu, acetate, propionate, and T-VFA, and negatively correlated (*p* < 0.05) with LDL-C. *g_Bacillus* was negatively correlated (*p* < 0.05) with DMI, Glu, acetate, propionate, butyrate, and T-VFA, but positively correlated (*p* < 0.05) with rumen pH and LDL-C. In summary, the DMI, nutrient digestibility, rumen fermentation, and serum parameters changed during the perinatal period of ewes, and the changes in DMI, serum glucose, acetate, propionate, and T-VFA were related to the rumen bacteria.

## 1. Introduction

Ewes undergo various physiological changes during the perinatal period, such as gestation and lactation, and they face severe challenges in nutrient digestion and energy balance [1]. The rapid development of the fetus in the late gestation period significantly increases the maternal demand for nutrients. Energy requirements rise by about 300%, and calcium requirements increase by more than 65% to support milk production in early lactation [2]. However, due to the regulatory role of the neuroendocrine network, the ewe’s physiological metabolism undergoes significant changes. The sharp decline in dry matter intake (DMI) during the perinatal period induces a negative energy balance (NEB), which clinically manifests as a reduction in serum glucose concentration [3,4]. Lipid mobilization can alleviate NEB to a certain extent. However, the non-esterified fatty acids (NEFAs) produced exceed the liver’s utilization capacity and are not completely oxidized into ketone bodies. Among these ketone bodies, β-hydroxybutyric acid (BHBA) constitutes approximately 80% [5]. Esposito et al. summarized the interactions between immunity, endocrinology, and metabolism in dairy cows during the transition period, and they showed that elevated concentrations of NEFA and BHBA in the blood to a certain degree trigger the immunosuppressive release of cytokines that induce inflammation, and inflammatory factors can negatively regulate DMI, further aggravating the metabolic disorders [6]. Moreover, the reduction in DMI will also affect the fermentation ability of rumen microbiota [7].

The energy source of ruminants mainly depends on the digestion and absorption of nutrients by rumen microbiota in fermented feed [3,8]. Short-chain fatty acids (SCFAs) produced by rumen microbial fermentation, such as propionate, are the substrates for gluconeogenesis and provide approximately 70% of the body’s energy [9]. Bacteroidetes, Firmicutes, and Proteobacteria are relatively abundant in the rumen [10]; it mainly includes cellulose-decomposing bacteria (*Fibrobacter succinogenes*, *Ruminococcus flavefaciens*, *Ruminococcus albus*, and *Butyrivibrio fibrisolvens*) [11] and hemicellulose digesting bacteria (*Prevotella*, *B. fibrisolvens*, and *R. flavefaciens*), which are present in almost all ruminants and can therefore be considered the core rumen bacterial microbiota [12]. Rumen microbiota profiles have been confirmed to be heritable and repairable, and there is a close relationship between them and animal production performance [13]. The changes in rumen microbiota are affected by many factors such as feed type, physiological state, and age [14]. For example, changes in rumen microbiota composition and abundance before and after delivery affect the production and distribution of SCFAs [2]. Interestingly, these changes may be due to the fact that feeding a low-fiber, high-energy diet after delivery increases the abundance of starch-degrading bacteria in the rumen [15]. A report indicated that production performance measures such as milk yield and milk composition are highly correlated with the abundance of various bacterial members in the rumen microbiota [16], which may therefore make the rumen microbiota composition an important regulator of ruminant production performance.

Although perinatal metabolism and gastrointestinal microbial changes have been described to some extent in dairy cows [17], different ways such as oral administration of zeolite were used to enhance energy metabolism and reproductive health in advanced gestation and the postpartum period [18]. However, dynamic changes in feed utilization, rumen fermentation, and serum biochemical indexes in perinatal ewes have not been systematically reported. Furthermore, there is a knowledge gap regarding the correlation between rumen microbiota and DMI, rumen fermentation parameters, and serum biochemical markers. Therefore, Hu sheep were used as the experimental animals in this study to explore the dynamic changes in DMI, nutrient digestibility, rumen fermentation parameters, and serum biochemical indexes in perinatal ewes as well as their relationship with the rumen microbiota.

## 2. Materials and Methods

### 2.1. Animals and Management

The donor animals and experimental procedures were approved by the requirements of the Ethical Committee of Hebei Agricultural University. In this study, all ewes (Hu sheep) were housed under the same breeding conditions in Lanhai Animal Husbandry Co., Ltd. (Zhangjiakou, China). Samples were collected from September to November 2021. Ten healthy late-gestation ewes with a body weight (BW) of 55.8 ± 5.09 kg and a body condition score (BCS) of 2.8 ± 0.27 at second parity with a similar day of gestation, as well as carrying twins (litter size was determined by transabdominal ultrasonography, HS-1600 V–7.5 MHz, Honda Electronics Co., Ltd., Toyohashi, Japan), were selected to study their dynamic development from the 120th day of gestation to the 15th day of lactation. Ewes were fed separately with one ewe in a single barn (2.0 × 1.3 m^2^), and each ewe was an independent experimental unit. The different time points during the perinatal period, such as Q21, Q14, and Q7 (21, 14, and 7 days before lambing), as well as H3, H7, and H14 (3, 7, and 14 days after lambing), were considered as different treatments according to the Repeated Measurement Model. Hence, the changes in associated parameters at six time points were obtained for each ewe, and the significance of differences at each time point among the 10 replicates was statistically analyzed. 

Ewes enrolled in this study were clinically healthy, and no symptoms of ketosis, abomasal displacement, or acute mastitis were observed. The experimental ewes were fed twice daily (8:00 a.m. and 5:00 p.m.) with the same total mixed ration (TMR) according to the recommendations of the feeding standard for sheep [19], and the same diet was given during late gestation and early lactation. All experimental animals were fed ad libitum, had free access to clean water, and showed approximately 5% feed refusal. The experiment period was divided into two phases: a 10-day adjustment period (pregnant ewes were acclimated to the facility and basal diet) and a 35-day feeding trial. Meanwhile, the daily feed supply and orts (i.e., refused feed) for each sheep were recorded to calculate the DMI. The dietary composition and nutritional levels are shown in Table 1.

### 2.2. Sample Collection and Chemical Analysis

Ten ewes were selected for a digestion and metabolism experiment to determine the digestibility of nutrients. During the experimental period, these ten ewes participated in six timed experiments, and feed and fecal samples (approximately 200 g) were collected on days Q21, Q14, Q7, H3, H7, and H14. Feed samples were collected at 08:00 daily. Ewes were fitted with fecal collection bags, and a harness was used for fecal bag attachment. Fecal samples were collected four times daily (06:00, 12:00, 18:00 and 24:00) for three consecutive days. A 10 mL aliquot of 10% hydrochloric acid (2.877 M) was added to the fresh feces from each ewe to prevent ammonia volatilization, and the samples were stored at −20 °C. At the end of the digestion and metabolism experiment, the feed and fecal samples were dried in a forced-air oven at 65 °C for 72 h, then weighed and milled through a 1 mm screen. Feed and fecal samples were analyzed for their dry matter (DM) (method 930.15), crude protein (CP) (method 984.13), ether extract (EE) (method 973.18), calcium (Ca) (method 927.02), and phosphorus (P) (method 984.27) contents, which were determined according to the indicated methods of AOAC [20]. The neutral detergent fiber (NDF) and acid detergent fiber (ADF) contents were analyzed by the method of Van Soest [21], using filter bags and fiber analysis equipment (Ankom A200; Ankom Technology, Macedon, NY, USA), and gross energy (GE) (Model 6300) was analyzed with an Automatic Isoperibol Oxygen Bomb Calorimeter (Parr Instruments, Moline, IL, USA).

Blood and rumen fluid were collected on days Q21, Q14, Q7, H3, H7, and H14 before the morning feeding. Blood samples were obtained from ewes by jugular venipuncture, which were allowed to coagulate, and the serum obtained by centrifugation at 3000× *g* for 15 min at 4 °C was stored at −20 °C prior to analysis. The serum glucose (Glu), β-hydroxybutyric acid (BHBA), non-esterified fatty acid (NEFA), triglyceride (TG), total cholesterol (TC), high-density lipoprotein cholesterol (HDL-C), low-density lipoprotein cholesterol (LDL-C), calcium (Ca), and phosphorus (P) concentrations were tested by colorimetric methods using commercial kits (Kaminuo Biology Co., Nanjing, China).

Rumen fluid samples were collected using an oral stomach tube, and fresh warm water and distilled water were used to clean the oral stomach tube thoroughly during sample collection. To avoid salivary contamination, the initial 30 mL of rumen fluid was discarded and the subsequent 60 mL was retained [22,23,24]. One 10 mL sample of filtrate from each ewe was stored in a 15 mL sterile tube with 2 mL of 25% HPO_3_ (metaphosphoric acid) for analysis of the volatile fatty acids (VFAs), and the other 10 mL sample of filtrate in a 15 mL sterile tube with 2 mL of 1% (w/v) H_2_SO_4_ (sulfuric acid) was used for the analyses of ammoniacal nitrogen (NH_3_-N) and microbial crude protein (MCP). All samples were stored at −20 °C until analysis. An additional 4 mL of rumen fluid was collected from each ewe and subsampled into two 2 mL sterile tubes, flash-frozen in liquid nitrogen, and stored at −80 °C until they were used for rumen microbial community analysis. Rumen fluid samples were immediately tested with an electronic pH meter (PHS-3C; Nanjing Nanda Analytical Instrument Application Research Institute, Nanjing, China). The concentration of NH_3_-N was measured by a UV spectrophotometer (UV1100; Shanghai Tyco Instruments Co., Shanghai, China). The concentration of MCP was determined by the Coomassie brilliant blue method. Rumen VFA concentrations were determined by gas chromatography (Varian 450, Agilent Technologies China, Co., Ltd., Beijing, China) using 2-ethylbutyric acid as an internal standard.

### 2.3. DNA Extraction and Sequencing

Ewes (*n* = 6) were randomly selected for DNA extraction and sequencing. The Power Soil DNA Isolation Kit (MoBio Laboratories, Carlsbad, CA, USA) was used to extract DNA from the samples according to the instructions. The integrity and concentration of genomic DNA were determined by 1% agarose gel and NanoDrop 2000 spectrophotometer (Thermo Fisher Scientific, Shanghai, China) analyses. The universal prokaryotic primers modified 341F (5′-CCTAYGGGRBGCASCAG-3′) and modified 806R (5′-GGACTACNNGGGTATCTAAT-3′) were used to amplify the V3-V4 hypervariable region of the 16S rRNA gene. To distinguish different samples, a 6 bp label sequence was added to the 5′ ends of the upstream and downstream primers (provided by Allwegene Company, Beijing, China). PCR was performed at Mastercycler Gradient (Eppendorf, Hamburg, Germany) with a total volume of 25 µL, including 12.5 µL KAPA 2G Robust Hot Start Ready Mix, 1 µL Forward Primer (5 µM), 1 µL Reverse Primer (5 µM), 5 µL DNA (6 ng/µL), and 5.5 µL H_2_O. The PCR reaction parameters were as follows: pre-denaturation for 5 min (95 °C), denaturation for 30 s (95 °C), annealing for 30 s (58 °C), extension for 30 s (72 °C) for 30 cycles, then extension for 5 min (72 °C), and finally storage at 4 °C. The PCR products were extracted by 2% agarose gel electrophoresis at a voltage of 80 v for 40 min. The PCR products were purified using the QIA Quick Gel Extraction Kit (QIAGEN, Hilden, Germany). The concentrations of PCR products after purification were determined using a NanoDrop 2000 spectrophotometer (Thermo Fisher Scientific), if it met the library standard. Subsequently, the amplified sequencing library was constructed and the 16S rRNA gene was sequenced using Allwegene Company’s (Beijing, China) Illumina Miseq PE 300 (Illumina, San Diego, CA, USA) platform.

### 2.4. Bioinformatics Analysis

The QIIME program (V1.9.1) was used to screen the original data to obtain reliable data. Pear (V0.9.6) software was used to filter and splice the data. Sequences with quality scores below 20 and lengths shorter than 225 bp were removed by double-ended splicing and chimera removal. When splicing, the minimum overlap was set to 10 bp, and the mismatch rate was 0.1. Clean sequences with an overlap longer than 10 bp were assembled using FLASH-1.2.11 and reads that could not be assembled were discarded. Chimera sequences were detected using usearch6.1. Then, these sequences were clustered with 97% similarity as operational taxonomic units (OTUs) [25]. Finally, the raw tags, clean tags, and clean tags for each individual and OTUs were obtained for days Q21, Q14, Q7, H3, H7, and H14 (Appendix A). The Chao 1 index and Shannon index reflecting alpha diversity were calculated using the Quantitative Insights into Microbial Ecology (QIIME) pipeline software (version 1.8.0), and partial least squares discriminant analysis (PLS-DA) based on OTU was used to reflect beta diversity. 

### 2.5. Statistical Analyses

This experiment was conducted in a completely randomized experimental design, with each ewe as the experimental unit. All data were tested for independence, normal distribution, and homogeneity using SPSS 22.0 (SPSS Inc., Chicago, IL, USA), and then evaluated using one-way ANOVA. Duncan’s multiple comparison method was used to evaluate the differences between groups, and *p* < 0.05 was considered to indicate a significant difference. Based on the random forest machine learning algorithm in R software (R 4.2.2 version), the mean-square error increase method was used to predict DMI and Glu through the rumen bacterial communities, and the taxa affecting perinatal DMI and Glu were obtained. These taxa were ranked according to the importance of features, and the number of taxa was determined by a cross-validation error curve. The common rumen bacterial genera affecting DMI and Glu were analyzed using a Venn diagram. A heatmap showed the changes in the relative abundance of DMI and Glu co-biomarkers during the perinatal period, and the correlations among perinatal microbial genera, feed digestibility, rumen fermentation parameters and serum biochemical indexes of ewes were analyzed using the Pearson correlation test.

## 3. Results

### 3.1. Changes in DMI and Nutrient Digestibility during the Perinatal Period of Ewes

The changes in DMI and nutrient digestibility of perinatal ewes are shown in Table 2. The DMI gradually decreased (*p* < 0.05) before lambing and increased (*p* < 0.05) after lambing. DMD, CPD, and ADFD gradually increased (*p* < 0.05), and there were no significant differences (*p* > 0.05) in EED, NDFD, and CaD before lambing, but these parameters decreased (*p* < 0.05) on day H3 and then increased (*p* < 0.05) on day H14. PD gradually increased (*p* < 0.05) before lambing and there was no difference (*p >* 0.05) after lambing.

### 3.2. Changes in Rumen Fluid Parameters during the Perinatal Period of Ewes

The changes in rumen fluid parameters of perinatal ewes are shown in Table 3. Rumen pH and NH_3_-N gradually increased (*p* < 0.05) before lambing, and then decreased (*p* < 0.05) after lambing compared to those on day Q7. The concentrations of acetate, butyrate, and T-VFA gradually decreased (*p* < 0.05) before lambing and then gradually increased (*p* < 0.05) after lambing. The propionate, MCP, and A/P had no differences (*p >* 0.05) before lambing, and propionate and MCP gradually increased (*p* < 0.05) after lambing, while A/P was lower (*p* < 0.05) on day H14 than on day Q21.

### 3.3. Changes in Serum Biochemical Indexes during the Perinatal Period of Ewes

The serum biochemical indexes at six sampling time points during the perinatal period are presented in Table 4. The concentration of Glu gradually decreased (*p* < 0.05) before lambing and increased (*p* < 0.05) after lambing. There were no differences (*p* > 0.05) from days Q14 to H14 in BHBA and NEFA concentrations and were higher (*p* < 0.05) than those on day Q21. The concentration of TG gradually increased (*p* < 0.05) before lambing and then decreased (*p* < 0.05) after lambing compared to that on day Q7. The concentration of TC gradually decreased (*p* < 0.05) during the perinatal period. There was no difference (*p* > 0.05) in LDL-C concentration before lambing, but it decreased (*p* < 0.05) after lambing. Regarding the concentration of Ca, no difference (*p* > 0.05) occurred from days Q7 to H7 and was higher (*p* < 0.05) than that on days Q21 and Q14. There were no significant differences (*p* > 0.05) in HDL-C and P concentrations between days Q14 and H14. However, these concentrations were significantly higher (*p* < 0.05) on day Q21 compared to days H3 and H14.

### 3.4. Correlation of the Rumen Microbiota with DMI, Nutrient Digestibility, Rumen Fermentation Parameters, and Serum Biochemical Indices during the Perinatal Period of Ewes

We collected rumen fluid samples at six time points before and after lambing and used amplicon sequencing to investigate the changes in rumen bacterial composition. The multi-sample rarefaction curve indicated the curve reached a plateau at about 30,000 reads, and the sequencing coverage was saturated (Figure 1A). There were no differences (*p* > 0.05) among the groups in Chao 1 and Shannon indexes (Figure 1B,C). PLS-DA analysis showed that bacterial community diversity changed along principal coordinates PC1 and PC2, and there were significant differences (*p* < 0.05) between pre- and postnatal samples (Figure 1D). The correlation among DMI, nutrient digestibility, rumen fermentation parameters, and serum biochemical indices showed that the DMI and Glu were positively correlated (*p* < 0.01) with acetate, propionate, butyrate, and T-VFA, but negatively correlated (*p* < 0.01) with LDL-C, TC, and pH (Appendix A). We observed that 24 rumen bacterial genera could explain the DMI changes of perinatal ewes (Figure 1E), and 18 rumen bacterial genera were identified as biomarkers for perinatal Glu (Figure 1F) changes when the cross-validation curve stabilized. The rumen bacterial genera related to DMI and Glu were predicted by random forest and analyzed using a Venn diagram, and six common rumen bacterial genera were observed (Appendix A). *g_Anaerovibrio*, *g_Schwartzia*, *g_Lachnobacterium*, and *g_Clostridium_sensu_stricto_3* gradually decreased before lambing and gradually increased after lambing. *g_Anaerovorax* gradually increased during the perinatal period. *g_Bacillus* gradually increased before lambing and gradually decreased after lambing (Figure 1G). The relative abundance of *g_Anaerovibrio*, *g_Lachnobacterium*, and *g_Schwartzia* was positively correlated (*p* < 0.05) with DMI, Glu, acetate, propionate, and T-VFA, but negatively correlated (*p* < 0.05) with LDL-C. *g_Bacillus* was negatively correlated (*p* < 0.05) with DMI, Glu, acetate, propionate, butyrate, and T-VFA, but positively correlated (*p* < 0.05) with pH and LDL-C (Figure 1H).

## 4. Discussion

The DMI is an important index for evaluating feed efficiency, which is related to production performance. High feeding capacity is the key to allowing ewes to pass the perinatal period smoothly [26]. A previous study found that the DMI in Hu sheep increased on days 40–100 of gestation and then gradually declined until lambing [27]. We also observed a reduction in DMI before lambing and an increase after lambing due to greater fetal growth at 30 days before lambing, which compressed the rumen volume [28]. And the digestibility of nutrients in Hu sheep pregnant with twins increased with a decrease in the diet level when they were restricted to feeding at various stages of gestation [27]. The passage rate of feed in the gastrointestinal tract is positively correlated with DMI [29], the slower the passage rate of ingested feed, the higher the digestibility of the diet, and the reduction in digestibility after lambing was related to increased feed intake, which improved the passage rate of feed in the rumen. In our study, nutrient digestibility was increased before lambing and decreased on day 3 after lambing which could be due to the change in DMI before lambing, the increase in nutrient digestibility on day 14 after lambing may be the result of the increased energy requirements of the ewes for lactation. The normal range of the rumen pH is 5.0~7.5, which is related to rumen metabolism, diet type, and fermentation pattern [30]. We observed that the rumen pH values of ewes before and after lambing were in the normal range, but the rumen pH after lambing was lower than before lambing, which was related to T-VFA. At the same time, we found that the concentrations of acetate, propionate, and butyrate decreased before lambing and increased after lambing, which may be related to the recovery of feed intake. NH_3_-N is an intermediate product of the breakdown of nitrogen-containing nutrients by rumen microbiota that can be used to synthesize MCP, and MCP is the most important nitrogen source for ruminants, providing 50–80% of their protein requirements [31]. In our study, the NH_3_-N concentration increased before lambing, while MCP did not show significant changes, indicating that although the rumen microbiota’s ability to produce ammonia increased, its capacity to synthesize bacterial proteins from ammonia remained unchanged in late pregnancy. However, the rumen A/P was decreased during the perinatal period, indicating a gradual transition from the acetate to propionate fermentation type. This suggests that more propionate can be used to produce glucose after lambing, which is conducive to energy supply and body condition recovery [32].

The changes in serum biochemical parameters can provide information on nutrient digestion, absorption, and utilization [33]. Glucose is a direct energy source for ruminants, mainly through the gluconeogenic synthesis of propionate produced by the rumen microbiota, and a small amount from intestinal degradation of non-fermentable carbohydrates [34]. Previous report indicated that the energy requirements of ewes increase by 180% in late pregnancy [35], while the reduction in DMI leads to a significant reduction in nutrient intake such as energy and protein, and the reduction in glucose concentration also reflects the NEB of ewes. In our study, we found that glucose concentration decreased before lambing and gradually increased after lambing which may be due to the fact that the increase in DMI promoted the production of volatile fatty acids in the rumen and thus increased the serum glucose content. Meanwhile, we also observed an increase in the concentrations of acetate, propionate, and butyrate after lambing. Lipid mobilization produces NEFA, which enters the liver for beta-oxidation under the action of carnitine palmitoyl transferase to produce acetyl-CoA. Acetyl-CoA binds to oxaloacetic acid and enters the tricarboxylic acid cycle (TCA) to completely oxidize and release ATP and supply energy [36], and NEFA can also promote the development of mammary glands and milk fat synthesis [37]. When the amount of oxaloacetic acid in the liver is insufficient, acetyl CoA is not completely oxidized to produce ketone bodies or esterified to TG. A previous study found that the reduction in DMI is negatively correlated with NEFA and BHBA [38]. However, in our study, we found that NEFA and BHBA were at high levels from 14 days before birth to 14 days after lambing, indicating that both DMI and Glu were increased after lambing, but there was still lipid mobilization in the body. The synthetic substrate of both cholesterol and BHBA is acetyl-CoA, and our study observed that the concentration of TC decreased in the perinatal period, which may be due to the inhibition of TC synthesis by the rise of BHBA [38,39]. Balthazar et al. found that fetuses in late pregnancy have higher requirements for trace elements such as calcium and phosphorus, and the contents of Ca and P in sheep milk are higher than in cow milk [40]. A serum total calcium concentration of ≤1.50 mmol/L is considered indicative of clinical hypocalcemia, while a concentration of ≤2.14 mmol/L is indicative of subclinical hypocalcemia [41]. In cows, serum Ca decreases at 9 h before calving and returns to the normal range about 72 h after calving [42]. Blood Ca levels at 24 h after calving were positively correlated with ketosis and metritis [41]. In this study, we found that the serum calcium concentration of ewes was higher after lambing, and the minimum concentration was more than 2.14 mmol/L, but no hypocalcemia occurred [43].

Feed intake and its chemical composition are the main factors determining the composition and function of the microbiota [44]. A previous study showed that changes in the rumen microbiota before and after lambing are mainly related to a high-starch diet [14]. Liu et al. found that the gut microbiota of empty-phase sows was more like that of pregnant sows in terms of α-diversity and taxonomic composition than that of lactating sows, which also emphasizes that the diets of pregnant sows and empty-phase sows were consistent, so the changes in the rumen microbiota could be attributed to the feed composition [45]. A previous study reported that different nutrition management strategies can promote changes in the rumen microbiota [46]. For example, the accepted feeding strategy is to switch to a relatively low-fiber, high-energy diet in postpartum dairy cows to maximize production performance, which increases the relative abundance of *Prevotella*, as well as decreases the fibrolytic bacteria (*Ruminococcus*, *Butyricum*, *Fibrobacter*, etc.) of Firmicutes after delivery [15,47]. These observations may be due to differences in dietary ingredients, nutritional value, and energy content affecting the relative abundance of rumen bacteria. In this study, the diet formulated according to NRC (2007) could meet the nutritional requirements of ewes in late gestation and early lactation, as well as removing the influence of dietary factors; we found that the β-diversity of rumen microbiota was different at different physiological stages in the perinatal period. Huang et al. analyzed the fecal microbiota of postpartum Holstein cows on day 1 and day 14, and found that there were more OTUs and the alpha diversity index was higher on day 1 after calving than on day 14 after calving [48]. In our study, there was no significant difference in α-diversity during the perinatal period, but there were different clustering patterns of β-diversity before lambing and after lambing. Therefore, in addition to dietary factors, changes in the perinatal physiological state and rumen metabolic pattern of ewes result in differences in rumen microbiota [49].

Multiple factors regulate the DMI in ruminants, and the influence of rumen microbiota on the energy requirement and feed degradation of ruminants has attracted much attention in recent years [48]. Our results indicated that DMI and Glu are positively correlated with acetate, propionate, butyrate, and T-VFA, but negatively correlated with LDL-C, TC, and pH. We speculate that the changes in rumen fermentation parameters and serum biochemical indices of ewes in the perinatal period might be attributed to the difference in DMI, affecting rumen fermentation and changing the gluconeogenesis rate to produce glucose, which may be related to the rumen microbiota. Based on the random forest algorithm, we predicted the rumen microbial genera that affect DMI and Glu changes in perinatal ewes, and *g_Anaerovibrio*, *g_Lachnobacterium*, *g_Schwartzia*, *g_Clostridium_sensu_stricto_3*, *g_Anaerovorax*, and *g_Bacillus* were common rumen biomarkers. *g_Anaerovibrio* encodes lipases that are involved in lipid synthesis and hydrolysis in ruminants [50]. *g_Lachnobacterium* is a Gram-positive rod anaerobic bacterium isolated from bovine rumen and feces, which mainly degrades glucose into lactate [51]. As a beneficial bacterium, it is also associated with a variety of diseases, and affects ovarian function by regulating changes in serum hormones [52]. The study has shown that *g_Lachnobacterium* can affect the secretion of glucagon-like peptide-1 (GLP-1) and regulate glucose homeostasis [53]. *g_Schwartzia* is a rumen bacterium that converts succinate into propionate, the only mechanism by which they produce energy [54]. Our study found that the *g_Anaerovibrio*, *g_Lachnobacterium,* and *g_Schwartzia* were positively correlated with DMI, glucose, acetate, propionate, butyrate, and T-VFA. With the increase in DMI, they made full use of various substrates fermentation to produce volatile fatty acids to provide energy for the body, thereby increasing serum glucose concentration. Ehling-Schulz et al. reported that *g_Bacillus* included a variety of pathogenic bacillus species, such as Bacillus cereus, which have few genes involved in the degradation of carbohydrate polymers, but contain a large number of degrading enzymes, cytotoxic proteins, and cell surface proteins, which were highly pathogenic in the gastrointestinal tract of animals [55]. It has also been found that Bacillus cereus can cause severe mammary infections in ruminants. Our study found that *g_Bacillus* was negatively correlated with DMI, Glu, acetate, propionate, butyrate, and T-VFA. This suggests that an increase in the relative abundance of *g_Bacillus* may reduce the function of the rumen, impair the immune system of ewes, and induce perinatal diseases.

## 5. Conclusions

In this study, we found that the DMI and glucose gradually decreased in late pregnancy and increased in early lactation. The digestibility of dry matter, crude protein, and acid detergent fiber gradually increased before lambing, and decreased on day 3 after lambing. The concentrations of acetate, propionate, butyrate, and T-VFA gradually decreased before lambing and increased after lambing. The rumen microbiota composition was different in perinatal ewes, and the changes in DMI, serum glucose, acetate, propionate, and T-VFA were related to rumen bacteria (*g_Anaerovibrio*, *g_Lachnobacterium*, *g_Schwartzia* and *g_Bacillus*). The results provide a basis for the regulation of physiological and biochemical parameters of perinatal ewes by rumen microbiota.

## Figures and Tables

**Figure 1 animals-14-02344-f001:**
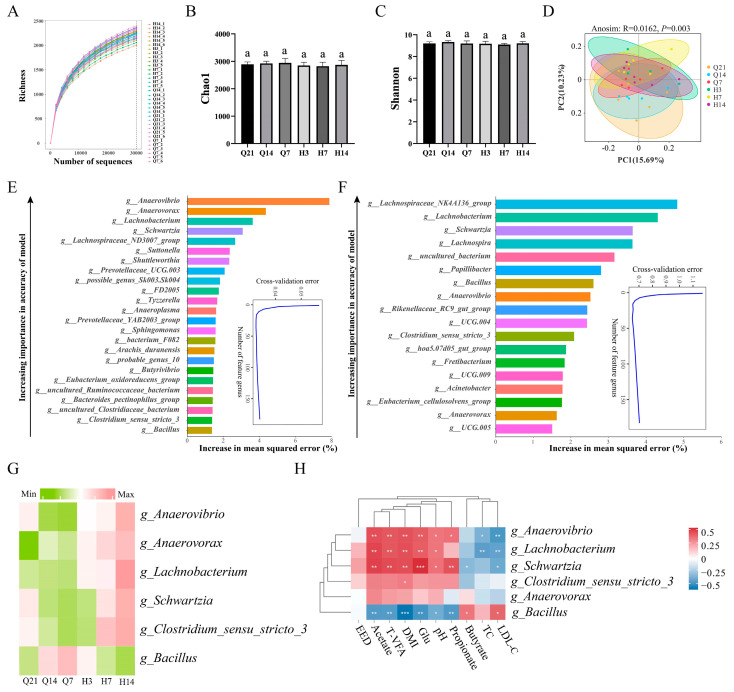
Correlation analysis of nutrient digestibility, fermentation parameters, blood biochemical indices, and the rumen microbiota in ewes during the perinatal period. (**A**) The rarefaction curve for each sample of ewes before and after lambing. The Chao1 (**B**) and Shannon index (**C**) at six sampling time points. (**D**) Partial least squares discriminant analysis based on the OTU for rumen bacteria in perinatal ewes (PLS-DA). Predictions of DMI (**E**) and serum Glu (**F**) by the rumen bacterial community based on a random forest model. The insert illustrates the results of a 50-fold cross-validation analysis. (**G**) Heatmap shows that the changes in the co-dominant bacterial genera of DMI and Glu during the perinatal period. (**H**) Correlation analysis between the DMI and Glu co-dominant bacterial genera and their significant correlation indicators. ^a^ Values within a row with different superscripts differ significantly at *p* < 0.05. * *p* < 0.05, ** *p* < 0.01, *** *p* < 0.001.

**Table 1 animals-14-02344-t001:** Diet composition and nutrient levels of the basal diet (dry matter basis).

Item	Content (%)
Ingredient	
Corn stalk silage	38.46
Peanut vine	15.38
Green hay	15.38
Corn	14.38
Bran	3.13
Soybean meal	11.65
CaHPO_4_	0.13
Premix ^(1)^	0.67
Sodium bicarbonate	0.40
NaCl	0.42
Total	100.00
Nutritional Indicator	
Metabolic energy, ME/(MJ kg^−1^) ^(2)^	9.40
Crude protein, CP	13.49
Ether extract, EE	2.60
Neutral detergent fiber, NDF	43.16
Acid detergent fiber, ADF	18.91
Crude ash	11.43
Calcium, Ca	0.66
Phosphorus, P	0.33

^(1)^ Provided per kilogram of diet (dry matter): Vitamin A 4402 IU, Vitamin D 755 IU, Vitamin E 126 IU, Cu 12.50 mg, Mn 28.30 mg, Zn 37.74 mg, Fe 40.88 mg, Co 0.85 mg, I 0.97 mg, and Se 0.85 mg. ^(2)^ ME is a calculated value, other nutritional indicators are measured values.

**Table 2 animals-14-02344-t002:** Dynamic changes in DMI and nutrient digestibility during the perinatal period of ewes.

Item ^(1)^	Prepartum			Postpartum			SEM	*p*-Value
	Q21	Q14	Q7	H3	H7	H14		
DMI (Kg/d)	1.53 ^b^	1.43 ^c^	1.39 ^c^	1.55 ^b^	1.82 ^a^	1.88 ^a^	0.031	<0.001
DMD	68.86 ^bc^	70.19 ^b^	74.27 ^a^	62.52 ^c^	65.75 ^c^	70.94 ^ab^	0.789	<0.001
CPD	69.77 ^b^	74.04 ^ab^	76.58 ^a^	69.50 ^b^	65.09 ^b^	74.11 ^ab^	0.877	<0.001
EED	74.95 ^b^	78.44 ^b^	82.20 ^ab^	73.51 ^c^	82.23 ^ab^	83.40 ^a^	0.861	<0.001
NDFD	52.76 ^b^	54.48 ^b^	58.10 ^ab^	53.71 ^b^	52.86 ^b^	59.99 ^a^	0.752	0.011
ADFD	43.89 ^b^	49.15 ^a^	48.93 ^a^	46.54 ^ab^	47.67 ^ab^	47.41 ^a^	0.727	0.224
CaD	43.46 ^bc^	44.13 ^bc^	47.78 ^b^	39.41 ^c^	42.31 ^c^	53.07 ^a^	0.962	<0.001
PD	61.14 ^b^	64.58 ^b^	68.89 ^a^	59.92 ^b^	60.60 ^b^	63.66 ^b^	0.748	0.001

^(1)^ DMI, dry matter intake; DMD, dry matter digestibility; CPD, crude protein digestibility; EED, ether extract digestibility; NDFD, neutral detergent fiber digestibility; ADFD, acid detergent fiber digestibility; CaD, calcium digestibility; PD, phosphorus digestibility. ^a–c^ Values within a row with different superscripts differ significantly at *p* < 0.05.

**Table 3 animals-14-02344-t003:** Changes in ruminal fermentation parameters during the perinatal period of ewes.

Item ^(1)^	Prepartum			Postpartum			SEM	*p*-Value
	Q21	Q14	Q7	H3	H7	H14		
pH	6.53 ^b^	6.54 ^b^	6.75 ^a^	6.47 ^b^	6.33 ^c^	6.30 ^c^	0.026	<0.001
NH_3_-N (mg/dL)	6.30 ^c^	6.42 ^b^	8.53 ^a^	7.46 ^b^	6.75 ^b^	7.78 ^b^	0.178	<0.001
MCP (mg/dL)	42.15 ^ab^	41.56 ^ab^	38.59 ^b^	40.48 ^b^	40.68 ^b^	43.49 ^a^	0.411	0.012
Acetate (mmol/L)	30.79 ^c^	31.56 ^c^	26.65 ^d^	34.49 ^b^	42.46 ^a^	43.41 ^a^	1.362	<0.001
Propionate (mmol/L)	10.74 ^c^	11.91 ^bc^	9.42 ^c^	12.53 ^b^	16.63 ^a^	18.17 ^a^	0.685	<0.001
Butyrate (mmol/L)	4.62 ^b^	4.18 ^b^	3.55 ^c^	4.05 ^b^	5.12 ^a^	5.60 ^a^	0.165	<0.001
A/P	2.88 ^a^	2.64 ^ab^	2.50 ^ab^	2.77 ^ab^	2.56 ^ab^	2.40 ^b^	0.064	0.129
T-VFA (mmol/L)	46.45 ^c^	45.65 ^c^	34.62 ^d^	51.07 ^b^	64.69 ^a^	66.71 ^a^	2.113	<0.001

^(1)^ NH_3_-N, ammoniacal nitrogen; MCP, microbial crude protein; T-VFAs, total volatile fatty acids; A/P, acetate to propionate ratio. ^a–d^ Values within a row with different superscripts differ significantly at *p* < 0.05.

**Table 4 animals-14-02344-t004:** Changes in serum biochemical indexes during the perinatal period of ewes.

Item ^(1)^	Prepartum			Postpartum			SEM	*p*-Value
	Q21	Q14	Q7	H3	H7	H14		
Glu (mmol/L)	2.14 ^b^	1.49 ^c^	1.42 ^c^	1.66 ^c^	2.63 ^a^	3.01 ^a^	0.120	<0.001
BHBA (mmol/L)	0.56 ^b^	0.61 ^a^	0.68 ^a^	0.64 ^a^	0.66 ^a^	0.81 ^a^	0.027	0.439
NEFA (mmol/L)	0.28 ^b^	0.53 ^a^	0.59 ^a^	0.57 ^a^	0.59 ^a^	0.72 ^a^	0.041	0.064
TG (mmol/L)	0.35 ^b^	0.42 ^b^	0.50 ^a^	0.44 ^ab^	0.38 ^b^	0.36 ^b^	0.107	0.179
TC (mmol/L)	2.28 ^a^	2.18 ^a^	1.74 ^b^	1.49 ^bc^	1.17 ^c^	1.17 ^c^	0.094	<0.001
HDL-C (mmol/L)	1.01 ^a^	0.84 ^ab^	0.63 ^b^	0.63 ^b^	0.62 ^b^	0.55 ^b^	0.034	0.002
LDL-C (mmol/L)	0.73 ^a^	0.75 ^a^	0.77 ^a^	0.59 ^ab^	0.38 ^b^	0.36 ^b^	0.047	0.015
Ca (mmol/L)	2.28 ^b^	2.25 ^b^	2.51 ^a^	2.50 ^a^	2.51 ^a^	2.38 ^ab^	0.024	<0.001
P (mmol/L)	2.24 ^a^	1.94 ^b^	2.09 ^ab^	1.96 ^b^	2.19 ^ab^	1.84 ^b^	0.043	0.056

^(1)^ Glu, glucose; BHBA, β-hydroxybutyric acid; NEFAs, non-esterified fatty acids; TGs, triglycerides; TC, total cholesterol; HDL-C, high-density lipoprotein cholesterol; LDL-C, low-density lipoprotein cholesterol; Ca, calcium; P, phosphorus. ^a–c^ Values within a row with different superscripts differ significantly at *p* < 0.05.

## Data Availability

The data sets generated in the current study are available in the Genome Sequence Archive repository (http://gsa.big.ac.cn/ accessed on 14 March 2024), under accession numbers CRA015355.

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
