# Peer review of "Dynamic Changes in the Nutrient Digestibility, Rumen Fermentation, Serum Parameters of Perinatal Ewes and Their Relationship with Rumen Microbiota"

_animals, 2024, doi:10.3390/ani14162344_

Round 1

Reviewer 1 Report

Comments and Suggestions for Authors

Dear Authors,

Congratulations on your work and the topic you have chosen. The transition period is a challenging phase across all species and merits thorough investigation to optimize and improve zoo-sanitary conditions, thereby facilitating a smoother transition for females. The current article is well-written, well-structured, and addresses an interesting and consistently relevant subject in sheep herds. A few minor revisions are listed below.

Line 19 "we" please correct with "We"

Line 36 "and were higher" sound better

Line 55-57 Rewrite this sentence "The rapid development of the fetus in the late gestation period increases the maternal demand for nutrients, such as the energy requirements which increase by 56 about 300% and calcium requirements which increase by more than 65% to support milk 57 production in early lactation" with "The rapid development of the fetus in the late gestation period significantly increases the maternal demand for nutrients. Specifically, energy requirements rise by about 300%, and calcium requirements increase by more than 65% to support milk production in early lactation."

Line 59 "changes a lot" sound better or maybe you can rewrite all sentence like this "However, due to the regulatory role of the neuroendocrine network, the ewe’s physiological metabolism undergoes significant changes. The sharp decline in dry matter intake (DMI) during the perinatal period induces a negative energy balance (NEB), which clinically manifests as a reduction in serum glucose concentration .

line 62-64 sound better someting like this "Lipid mobilization can alleviate NEB to a certain extent. However, the non-esterified fatty acids (NEFAs) produced exceed the liver's utilization capacity and are not completely oxidized into ketone bodies. Among these ketone bodies, β-hydroxybutyric acid (BHBA) constitutes approximately 80%. please rewrite

Line 84, here maybe you can also specify that in cows different ways were used to improve the transition period, I let here a study that made this "The effect of oral administration of zeolite on the energy metabolism and reproductive health of Romanian spotted breed in advanced gestation and post partum period" maybe you could cite him 

In the introduction section, it would be necessary to include a paragraph about the species of ruminal bacteria and their activities. Additionally, the introduction should conclude with a sentence clearly stating the purpose of the study.

Line 111 "5 pm." if you say 17:00 it's not necessary pm

Line 125 remove 1 space between sentences

The materials and methods section is well-written and allows for replication of the study. However, I have one more question: Among the 10 females, were there any animals with twin or triplet births? If not, was this purely incidental, or were such animals excluded beforehand?

Line 249 "but were higher" sound better

Line 253 Regarding the concentration

Line 255-256 rewrite with this "There were no significant differences (p>0.05) in HDL-C and P concentrations between days Q14 and H14. However, these concentrations were significantly higher (p<0.05) on day Q21 compared to days H3 and H14."

Line 342-346 The phrase is too long and difficult to understand, you need to rewrite, maybe will be useful to split her 

Line 357-359 replace with A serum total calcium concentration of ≤1.50 mmol/L is considered indicative of clinical hypocalcemia, while a concentration of ≤2.14 mmol/L is indicative of subclinical hypocalcemia.

The results, discussions, and conclusions are well-written and easily understood by the reader.

Comments on the Quality of English Language

 Moderate editing of English language required

Author Response

Response to reviewer 1 comments

We would like to express sincere appreciation to you and the anonymous reviewers who had given these constructive comments and suggestions. These comments and suggestions are all valuable and very helpful for revising and improving our manuscript entitled “Dynamic changes in the nutrient digestibility, rumen fermentation, serum parameters of perinatal ewes and relationship with rumen microbiota” (ID: animals-3153169), as well as the important guiding significance to our researches. According to the manuscript decision from the editorial office, we have substantially revised the manuscript considering the comments and suggestions. Detailed responses are given in the following parts.

Comments 1: Line 19 "we" please correct with "We"

Response 1: Thank you for pointing this out. We have revised the error and marked it with red in the manuscript. Revised content in Lines 20.

Comments 2: Line 36 "and were higher" sound better

Response 2: Thank you for pointing this out. We have revised the content "and higher" to "and were higher". Revised content in Lines 37.

Comments 3: Line 55-57 Rewrite this sentence "The rapid development of the fetus in the late gestation period increases the maternal demand for nutrients, such as the energy requirements which increase by 56 about 300% and calcium requirements which increase by more than 65% to support milk 57 production in early lactation" with "The rapid development of the fetus in the late gestation period significantly increases the maternal demand for nutrients. Specifically, energy requirements rise by about 300%, and calcium requirements increase by more than 65% to support milk production in early lactation."

Response 3: Thank you for pointing this out. We have revised the manuscript based on your comments. Revised content in Lines 59-62.

Comments 4: Line 59 "changes a lot" sound better or maybe you can rewrite all sentence like this "However, due to the regulatory role of the neuroendocrine network, the ewe’s physiological metabolism undergoes significant changes. The sharp decline in dry matter intake (DMI) during the perinatal period induces a negative energy balance (NEB), which clinically manifests as a reduction in serum glucose concentration.

Response 4: Thank you for pointing this out. We have revised the manuscript based on your comments. Revised content in Lines 66-69.

Comments 5: line 62-64 sound better something like this "Lipid mobilization can alleviate NEB to a certain extent. However, the non-esterified fatty acids (NEFAs) produced exceed the liver's utilization capacity and are not completely oxidized into ketone bodies. Among these ketone bodies, β-hydroxybutyric acid (BHBA) constitutes approximately 80%. please rewrite.

Response 5: Thank you for pointing this out. We have revised the manuscript based on your comments. Revised content in Lines 73-76.

Comments 6: Line 84, here maybe you can also specify that in cows different ways were used to improve the transition period, I let here a study that made this "The effect of oral administration of zeolite on the energy metabolism and reproductive health of Romanian spotted breed in advanced gestation and postpartum period" maybe you could cite him. 

Response 6: Thank you for pointing this out. We agree with this comment. We have revised it in the manuscript and cited this reference. Revised content in Lines 103-106.

Comments 7: In the introduction section, it would be necessary to include a paragraph about the species of ruminal bacteria and their activities. Additionally, the introduction should conclude with a sentence clearly stating the purpose of the study.

Response 7: Thank you for pointing this out. We have added the species of ruminal bacteria and their activities in the introduction, and the revised contents are as follows: Bacteroidetes, Firmicutes, and Proteobacteria are relatively abundant in the rumen [10], it mainly includes cellulose decomposing bacteria (Fibrobacter succinogenes, Rumino-coccus flavefaciens, Ruminococcus albus, and Butyrivibrio fibrisolvens) [11] and hemi-cellulose digesting bacteria (Prevotella, B. fibrisolvens, and R. flavefaciens), which are present in almost all ruminants and can therefore be considered the core rumen bacte-rial microbiota [12]. Rumen microbiota profiles have been confirmed to be heritable and repairable, and there is a close relationship between them and animal production performance [13]. Revised content in Lines 86-93.

We have revised the purpose of the study in lines 110-113. “Therefore, Hu sheep were used as the experimental animals in this study to explore the dynamic changes of DMI, nutrient digestibility, rumen fermentation parameters and serum biochemical indexes in perinatal ewes as well as their relationship with the rumen microbiota.”

[10] Xue, M. Y.; Sun, H. Z.; Wu, X. H.; Liu, J. X.; Guan, L. L. Multi-omics reveals that the rumen microbiome and its metabolome together with the host metabolome contribute to individualized dairy cow performance. Microbiome 2020, 8, doi:10.1186/s40168-020-00819-8.

[11] Krause, D.O.; Denman, S.E.; Mackie, R.I.; Morrison, M.; Rae, A.L.; Attwood, G.T.; et al. Opportunities to improve fiber degradation in the rumen: microbiology, ecology, and genomics. FEMS Microbiology Reviews 2003, 27, 663-93, doi:10.1016/S0168-6445(03)00072-X.

[12] Sarah, M.; Itzhak, M.; Islands in the stream: from individual to communal fiber degradation in the rumen ecosystem, FEMS Microbiology Reviews 2019, 4, doi:10.1093/femsre/fuz007.

[13] Zhao, X.; Zhang, Y.; Rahman, A.; Chen, M.; Li, N.; Wu, T.; et al. Rumen microbiota succession throughout the perinatal period and its association with postpartum production traits in dairy cows: A review, Animal Nutrition 2024, 18:17-26, doi: 10.1016/j.aninu.

Comments 8: Line 111 "5 pm." if you say 17:00 it's not necessary pm.

Response 8: Thank you for pointing this out. We have revised the manuscript according to your comments. Revised content in Lines 134.

Comments 9: Line 125 remove 1 space between sentences.

Response 9: Thank you for pointing this out. We have revised the manuscript according to your comments. Revised content in Line 149.

Comments 10: The materials and methods section is well-written and allows for replication of the study. However, I have one more question: Among the 10 females, were there any animals with twin or triplet births? If not, was this purely incidental, or were such animals excluded beforehand?

Response 10: We are sorry for your confusion. In this study, ten healthy late-gestation ewes with a body weight (BW) of 55.8±5.09 kg and a body condition score (BCS) of 2.8±0.27 at second parity with a similar day of gestation, as well as carrying twins (litter size was determined by transabdominal ultrasonography, HS-1600V-7.5MHz, Japan) were selected to study their dynamic development from the 120th day of gestation to the 15th day of lactation. Revised content in Line 122.

Comments 11: Line 249 "but were higher" sound better.

Response 11: Thank you for pointing this out. We agree with this comment. Therefore, we have revised the content "but higher" to "but were higher". Revised content in Line 277.

Comments 12: Line 253 Regarding the concentration.

Response 12: We are sorry for your confusion. Meanwhile, we have revised the content "and higher" to "and was higher". Revised content in Lines 284.

Comments 13: Line 255-256 rewrite with this "There were no significant differences (p>0.05) in HDL-C and P concentrations between days Q14 and H14. However, these concentrations were significantly higher (p<0.05) on day Q21 compared to days H3 and H14."

Response 13: Thank you for pointing this out. We have revised the manuscript according to your comments. Revised content in Lines 286-288.

Comments 14: Line 342-346 The phrase is too long and difficult to understand, you need to rewrite, maybe will be useful to split her. 

Response 14: Thank you for pointing this out. We have revised the manuscript according to your comments. Revised content in Lines 380-384.

Comments 15: Line 357-359 replace with A serum total calcium concentration of ≤1.50 mmol/L is considered indicative of clinical hypocalcemia, while a concentration of ≤2.14 mmol/L is indicative of subclinical hypocalcemia.

Response 15: Thank you for pointing this out. We have revised the manuscript according to your comments. Revised content in Lines 397-399.

Comments 16: Moderate editing of English language required.

Response 16: Thanks a lot for your careful and valuable comment. We polished the English language of the paper by the website (https://www.scribendi.com).

Reviewer 2 Report

Comments and Suggestions for Authors

The study investigates changes in dry matter intake (DMI), nutrient digestibility, rumen fluid parameters, and serum biochemical indices in perinatal ewes. The findings highlight the significant role of specific rumen bacteria in influencing physiological and biochemical parameters during the perinatal period. The paper provides valuable data on these changes, meriting publication.

I have two major concerns that need to be addressed by the authors in a revised version of the manuscript:

  1. Lines 112-113: “The same diet was given during late-gestation and early-lactation.” In sheep husbandry, the dry period (late-gestation) ration is recommended to be different from the early-lactation ration, at least in the Ca content. This inconsistency should be carefully reviewed by the authors in the discussion section.

  2. Line 151: “Rumen fluid samples were collected using an oral stomach tube.” Sampling rumen fluid by an oral stomach tube is questionable due to possible contamination by saliva. Discarding the initial 30 mL of rumen fluid (lines 153-154) does not fully address potential pH elevation from saliva contamination. The authors should provide references to confirm that their procedures guarantee accurate pH measurement. Note that the pH values of the rumen fluids in Table 3 are quite elevated.

In addition, several grammatical and clarity issues include:

  • Line 19: “we” should be “We”

  • Lines 25-26: Explain what “g” means.

  • Lines 62-63: Correct “NEFA” to “Non-esterified fatty acids”.

  • Lines 65-68: The statement about NEFA and BHBA's effects needs further references. Ketone bodies typically do not affect ruminants clinically, unlike reduced blood glucose.

  • Lines 99-100: Report the number of embryos the ewes carried.

  • Lines 110-112: Provide more details on feeding. Was the amount restricted? Was feeding ad libitum?

  • Table 1: Correct units of Metabolic Energy; erase “2)”

  • Lines 348-349: Provide references for the correlation between reduced DMI and NEFA/BHBA.

  • Line 48: Change “were related to rumen bacteria” to “were related to the rumen bacteria”.

  • Line 19: Change "serum biochemical indexes in perinatal ewes and relationship with rumen microbiota." to "serum biochemical indexes in perinatal ewes and their relationship with rumen microbiota."

  • Line 108: Change “could be statistically analyzed” to “was statistically analyzed”.

  • Line 130: “2.877 mol/L” should be “2.877 M”.

  • Lines 225-227: Rephrase to "There were no differences (p>0.05) in EED, NDFD, and CaD before lambing, but these parameters decreased (p<0.05) on day H3 and then increased (p<0.05) on day H14."

  • Lines 235-236: Rephrase to "Rumen pH and NH3-N gradually increased (p<0.05) before lambing, and then decreased (p<0.05) after lambing compared to those on day Q7."

  • Lines 252-253: Rephrase to "There was no difference (p>0.05) in LDL-C concentration before lambing, but it decreased (p<0.05) after lambing."

  • Lines 323-326: Correct to "The NH3-N concentration increased before lambing, while MCP did not show significant changes, indicating that although the rumen microbiota's ability to produce ammonia increased, its capacity to synthesize bacterial proteins from ammonia remained unchanged in late pregnancy."

Author Response

Response to reviewer 2 comments

We would like to express sincere appreciation to you and the anonymous reviewers who had given these constructive comments and suggestions. These comments and suggestions are all valuable and very helpful for revising and improving our manuscript entitled “Dynamic changes in the nutrient digestibility, rumen fermentation, serum parameters of perinatal ewes and relationship with rumen microbiota” (ID: animals-3153169), as well as the important guiding significance to our researches. According to the manuscript decision from the editorial office, we have substantially revised the manuscript considering the comments and suggestions. Detailed responses are given in the following parts.

Comments 1: Lines 112-113: “The same diet was given during late-gestation and early-lactation.” In sheep husbandry, the dry period (late-gestation) ration is recommended to be different from the early-lactation ration, at least in the Ca content. This inconsistency should be carefully reviewed by the authors in the discussion section.

Response 1: We are sorry for your confusion. There are two main considerations for feeding the same diet in late-gestation and early-lactation. On the one hand, we looked up the nutritional requirements of sheep and found that the dry matter, energy, protein, calcium and phosphorus requirements of ewes with weighing 55kg and carrying twins were similar in late-gestation and early-lactation, and the diet formulated according to NRC (2007) could meet the nutritional requirements including Ca of perinatal ewes. On the other hand, previous a study has reported that different nutrition management strategies can promote changes in rumen microbiota [1]. For example, the accepted feeding strategy is to switch a relatively low-fiber, high-energy diet in postpartum dairy cow to maximize production performance, which increases the relative abundance of Prevotella, as well as decrease the fibrolytic bacteria (Ruminococcus, Butyricum and Fibrobacter, etc.) of Firmicutes after delivery [2-3]. These observations may be due to differences in dietary ingredients, nutritional value, and energy content affecting the relative abundance of rumen bacteria. However, the main purpose of our study was to reduce the regulatory effect of dietary differences on rumen microbiota on the basis of meeting the nutritional requirements of ewes in the perinatal period, and to explore the changes of physiological and biochemical indicators of ewes under different physiological states and their relationship with rumen microbiota. Revised content in Lines 412-423.

  • Simon, D.; Camarinha-Silva, Amé; Conrad Jürgen, Uwe. B.; Markus, R.; Jana, S. A Structural and Functional Elucidation of the Rumen Microbiome Influenced by Various Diets and Microenvironments. Frontiers in Microbiology 2017, 8, 1605, doi:10.3389/fmicb.2017.01605.
  • ; López-García.; González-Recio.; Elcoso.; Fàbregas.; Chaucheyras-Durand.; et al. Changes in the rumen and colon microbiota and effects of live yeast dietary supplementation during the transition from the dry period to lactation of dairy cows. Journal of dairy science 2019, 102, 6180-6198, doi:10.3168/jds.2018-16105.
  • Lima, F.S.; Oikonomou, G.; Lima, S.F.; Bicalho, M.L.S.; Ganda, E.K.; de Oliveira Filho, J.C.; Lorenzo, G.; Trojacanec, P.; Bicalho, R.C. Prepartum and Postpartum Rumen Fluid Microbiomes: Characterization and Correlation with Production Traits in Dairy Cows. Applied and Environmental Microbiology 2015, 81, 1327–1337, doi:10.1128/AEM.03138-14.

Comments 2: Line 151: “Rumen fluid samples were collected using an oral stomach tube.” Sampling rumen fluid by an oral stomach tube is questionable due to possible contamination by saliva. Discarding the initial 30 mL of rumen fluid (lines 153-154) does not fully address potential pH elevation from saliva contamination. The authors should provide references to confirm that their procedures guarantee accurate pH measurement. Note that the pH values of the rumen fluids in Table 3 are quite elevated.

Response 2: We are sorry for your confusion. According to previous reports, oral intubation for rumen fluid collection has been widely used. The ruminal fluids were collected by the following method: One end of the prerinsed and sterilized sampler with a metal filter was put into the rumen, and then, a 50 ml syringe fixed at the other end was used to extract the rumen fluid, discarding the first tube of rumen fluid to avoid saliva contamination and saving the second tube of rumen fluid [1-3]. The insertion depth was considered, as the tube approximately reached the bottom of the rumen based on the body size of each animal. The ruminal pH value usually fluctuates between 5.5-7.5 [4]. Palmonari et al. recorded pH at 10-min intervals over a 54-h period and the PH range was 6.11 to 6.51[4]. When the pH is at 6.2-6.8, the rumen microbial activity is the strongest, in which most feed fiber can be effectively degraded, as well as both Ruminococcus albus and R. flavefaciens are unable to grow at pH <6.0 in continuous culture. Yun-Xia et al were also found no differences in pH (range 6.31-6.86) between different treatments [1]. Therefore, we found rumen PH range of 6.30 to 6.75 without salivary contamination. At the same time, we added references to the manuscript to support our rumen fluid collection operations. Revised content in Lines 177-178.

  • Yun-Xia, G.; Ruo-Chen, Y.; Chun-Hui, D.; Yong, W.; Qing-Hong, H.; Shou-Kun, J.; et al. Effect of Dioscorea Opposite Waste on Growth Performance, Blood Parameters, Rumen Fermentation and Rumen Bacterial Community in Weaned Lambs. Scientia Agricultura Sinica 2023, 22, 1833–1846, doi:10.1016/j.jia.2022.10.002.
  • Mao, Y.; Wang, F.; Kong, W.; Wang, R.; Liu, X.; Ding, H.; et al. Dynamic changes of rumen bacteria and their fermentative ability in high-producing dairy cows during the late perinatal period. Frontiers in Microbiology 2023, 25, 1269123, doi: 10.3389/fmicb.2023.1269123.
  • Gao, J.; Sun, Y.; Bao, Y.; Zhou, K.; Kong, D.; Zhao, G. Effects of different levels of rapeseed cake containing high glucosinolates in steer ration on rumen fermentation, nutrient digestibility and the rumen microbial community. British Journal of Nutrition 2021, 125, 266-274, doi: 10.1017/S0007114520002767.
  • Colman, E.; Fokkink, W.B.; Craninx, M.; Newbold, J.R.; De Baets, B.; Fievez, V. Effect of induction of subacute ruminal acidosis on milk fat profile and rumen parameters. Journal of Dairy Science 2010, 93, 4759-73, doi: 10.3168/jds.2010-3158.
  • Palmonari, A.; Stevenson, D.M.; Mertens, D.R.; Cruywagen, C.W.; Weimer, P.J. pH dynamics and bacterial community composition in the rumen of lactating dairy cows. Journal of Dairy Science 2010, 93, 279-87, doi: 10.3168/jds.2009-2207.

Comments 3: “we” should be “We”.

Response 3: Thank you for pointing this out. We agree with this comment. Therefore, we have revised the error and marked it in red in the manuscript. Revised content in Lines 20.

Comments 4: Lines 25-26: Explain what “g” means.

Response 4: We are sorry for your confusion. Microorganisms are divided into kingdoms, phyla, classes, orders, families, genera, and species according to biological classification. “g” means genera.

Comments 5: Lines 62-63: Correct “NEFA” to “Non-esterified fatty acids”.

Response 5: Thank you for pointing this out. We agree with this comment. Therefore, we have revised the error and marked it in red in the manuscript. Revised content in Lines 73-74.

Comments 6: Lines 65-68: The statement about NEFA and BHBA's effects needs further references. Ketone bodies typically do not affect ruminants clinically, unlike reduced blood glucose.

Response 6: Thank you for pointing this out. We have added new references to support the statement in the manuscript about the effects of NEFA and BHBA on ruminants. Revised content in Lines 76-81.

Comments 7: Lines 99-100: Report the number of embryos the ewes carried.

Response 7: Thank you for pointing this out. We have added the number of embryos the ewes carried. “Ten healthy late-gestation ewes with a body weight (BW) of 55.8±5.09 kg and a body condition score (BCS) of 2.8±0.27 at second parity with a similar day of gestation, as well as carrying twins (litter size was determined by transabdominal ultrasonography, HS-1600V-7.5MHz, Japan) were selected to study their dynamic development from the 120th day of gestation to the 15th day of lactation.” Revised content in Line 122.

Comments 8: Lines 110-112: Provide more details on feeding. Was the amount restricted? Was feeding ad libitum?

Response 8: We are sorry for your confusion. All experimental animals were feeding ad libitum and free to access clear water. Revised content in Lines 136-137.

Comments 9: Table 1: Correct units of Metabolic Energy; erase “2)”

Response 9: Thank you for pointing this out. We have revised the error and marked it in red in the manuscript.

Comments 10: Lines 348-349: Provide references for the correlation between reduced DMI and NEFA/BHBA.

Response 10: Thank you for pointing this out. We have added references for the correlation between reduced DMI and NEFA/BHBA. Revised content in Lines 387-388.

Comments 11: Line 48: Change “were related to rumen bacteria” to “were related to the rumen bacteria”.

Response 11: Thank you for pointing this out. We agree with this comment. We have revised “were related to rumen bacteria” to “were related to the rumen bacteria”. Revised content in Line 49.

Comments 12: Line 19: Change "serum biochemical indexes in perinatal ewes and relationship with rumen microbiota." to "serum biochemical indexes in perinatal ewes and their relationship with rumen microbiota."

Response 12: Thank you for pointing this out. We agree with this comment. We have revised "serum biochemical indexes in perinatal ewes and relationship with rumen microbiota." to "serum biochemical indexes in perinatal ewes and their relationship with rumen microbiota". Revised content in Line 19.

Comments 13: Line 108: Change “could be statistically analyzed” to “was statistically analyzed”.

Response 13: Thank you for pointing this out. We agree with this comment. We have revised “could be statistically analyzed” to “was statistically analyzed”. Revised content in Line 130.

Comments 14: Line 130: “2.877 mol/L” should be “2.877 M”.

Response 14: Thank you for pointing this out. We agree with this comment. We have revised “2.877 mol/L” should be “2.877 M”. Revised content in Line 154.

Comments 15: Lines 225-227: Rephrase to "There were no differences (p>0.05) in EED, NDFD, and CaD before lambing, but these parameters decreased (p<0.05) on day H3 and then increased (p<0.05) on day H14."

Response 15: Thank you for pointing this out. We agree with this comment. We have revised the content and marked it with red in Lines 251-253.

Comments 16: Lines 235-236: Rephrase to "Rumen pH and NH3-N gradually increased (p<0.05) before lambing, and then decreased (p<0.05) after lambing compared to those on day Q7."

Response 16: Thank you for pointing this out. We agree with this comment. We have revised the content and marked it with red in Lines 262-264.

Comments 17: Lines 252-253: Rephrase to "There was no difference (p>0.05) in LDL-C concentration before lambing, but it decreased (p<0.05) after lambing."

Response 17: Thank you for pointing this out. We agree with this comment. We have revised the content and marked it with red in Lines 282-283.

Comments 18: Lines 323-326: Correct to "The NH3-N concentration increased before lambing, while MCP did not show significant changes, indicating that although the rumen microbiota's ability to produce ammonia increased, its capacity to synthesize bacterial proteins from ammonia remained unchanged in late pregnancy."

Response 18: Thank you for pointing this out. We agree with this comment. We have revised the content and marked it in red in Lines 358-361.
